



# The role of dust sources in the Tibetan Plateau may be Underestimated: A high-resolution simulation study of regional differences in dust characteristics in the Tibetan Plateau

Yunshu Zhang[1], Jiening Liang[1], Zhida Zhang[1], Bentao Li[1], Haotian Zhang[1], Xianjie Cao[1], Pengfei Tian[1], Lei Zhang[1,2]

1 Key Laboratory of Semi-Arid Climate Change, Ministry of Education, College of Atmospheric Sciences, Lanzhou University, Lanzhou 730000, Gansu, China

Provincial-Ministry Collaborative Innovation Center for Western Ecological Security, Lanzhou University, Lanzhou
730000, Gansu, China

Correspondence to:   Lei Zhang (zhanglei@lzu.edu.cn)

**Keywords:** dust sources, the Tibetan Plateau, dust concentration, WRF-Chem

**Abstract:** Because of the unique geographical location of the Tibetan Plateau and its important role in global climate change, aerosol variability over the plateau has been of wide interest to the academic
community. Most studies have focused on the influence of external aerosols; however, a few studies have been conducted on dust aerosols within the plateau. In this study, the plateau was divided into three regions, west, south, and north based on surface vegetation and climatic characteristics, and the Weather Research and Forecasting model with Chemistry was used to simulate the Tibetan Plateau from 2004 to 2006 to quantify the spatial and temporal variability of dust within the plateau with high
resolution. The dust sources of the plateau are located in the northern part of the Qiangtang Plateau, the Yarlung Tsangpo River basin, the Namucuo and Lhasa regions, the Qaidam Basin, the source areas of the Yellow and Yangtze Rivers, and the Qinghai Lake and its surrounding areas. Owing to windy weather and arid soil conditions, the dust emissions of the three regions reached $11.00 \times 10^7$ (west), $3.30 \times 10^7$ (south) and $4.50 \times 10^7$ (north) $\mu g \cdot m^{-2}$, during winter, and remained at a low level from May
to October. Although the annual variation in dust emissions was relatively consistent across the three regions, there were substantial differences in dust loading, with almost no dust present in the atmosphere in the south, a peak dust loading of $94.00 \times 10^5$ $\mu g \cdot m^{-2}$ in January in the west, and a bimodal structure in the north with peaks in April and October and a maximum value of $13.00 \times 10^{10}$ $\mu g \cdot m^{-2}$, which was primarily influenced by the temperature 2 m above the ground. In summer 10% of the dust
that starts in the interior of the plateau can be transported to the upper troposphere (above 8 km).



## Introduction

Dust is one of the major components of atmospheric aerosols, with approximately $100\text{-}200 \times 10^6$ tons of dust is emitted into the atmosphere each year, accounting for almost half of the total aerosols in the troposphere (Shi and Zhao, 2003), and it can exert a substantial impact on climate, air quality and human health. The direct and indirect effects of dust can influence the energy budget of the Earth-atmosphere system (Tegen and Lacis, 1996; Huang et al., 2006b; Huang et al., 2006a). In addition, dust can act as cloud condensation nuclei, and the amount of dust can affect cloud volume, cloud duration, rainfall, and intensity (Huang et al., 2006b; Huang et al., 2006a; Han et al., 2009; Li et al., 2011). Moreover, dust can influence clouds and precipitation via its interactions with clouds (Liu et al., 2019b; Liu et al., 2019a). These properties of dust are of great interest to the academic community.

The Tibetan Plateau (TP), the highest plateau in the world, is known as the "roof of the world" and has a complex terrain with altitudes ranging from 1,500 to 5,000 meters. It is also known as the "water tower of Asia", where the Yangtze, Yellow, Lancang, Yarlung Tsangpo, Ganges, and Indus Rivers all originate. The complex topography and sub-surface conditions make the climate very different in different regions of the TP, with most areas southeast of the TP having a warm, humid climate with lush vegetation, whereas the Ali region west of the TP is extremely cold in winter and extremely dry with little rain because of the blockage of water vapor. Little vegetation grows on the surface of this area. The northern part of the TP is primarily a temperate climate zone, hot and rainy in summer and dry with little rain in winter (Lin, 1981).

There are many well-known dust source areas around the TP, such as the Taklamakan Desert, Thar Desert of India, Badain Jaran Desert, and Gurbantung Desert. To date, many scholars have made great progress in the study of dust transport mechanisms, routes, and even the amount of dust transported from different dust sources to the TP. Chen (Chen et al., 2013) studied a summer dust transport event from the Taklamakan Desert to the TP. The transport was further enhanced by the presence of a strong cold front system over the Taklamakan Desert combination with a weakened summer thermal effect and westerly winds. This caused the dust to break through the planetary boundary layer and extend into the upper troposphere in the northern part of the plateau and produced an average daily transport of 6.6 Gg of dust from the Taklamakan Desert to the TP over the 5 days of the event. Further studies have suggested that dust transported to the northern slope of the TP is partially driven eastward by northwesterly air currents and partially continues southward (Liu et al., 2015; Jia et al., 2015). The results of Wang (Wang et al., 2021) provide a mechanism for the transport of dust from India over the Himalayas to the highlands, where strong dust storms occur under the combined effect of the high altitude rapids, high altitude troughs, and subtropical high pressure over the Thar Desert in India. The typical north-south secondary circulation formed in the outlet region of



the high altitude rapids, causes the dust to rise more strongly to higher ground; at this point, southerly flow in front of the low-pressure trough over Afghanistan and the southern branch of the trough over Bangladesh readily carry these upwelling dust particles to the Himalayas. Hu (Hu et al., 2020) studied the total amount of dust transported to the TP from different dust sources at different altitudes, and dust from North Africa and East Asia was primarily transported from the northern part of the plateau to the plateau and from the Middle East to the plateau from the southern slope. The height at which dust transport is more concentrated is 3-6 km, and a total of 42.3 Tg of dust can be transported from the three regions to the TP on average per year.

After being transported to the TP, the dust is then moved to higher atmospheric altitudes because of the altitude of the TP and the heat pumping effects during summer. Distinct aerosol bands can be observed at the top of the troposphere over the plateau during the monsoon (Xu et al., 2018; Zhang et al., 2020a; Ma et al., 2019; Feng et al., 2020). Thereafter, they are transported outward; the dust emanating from the plateau delays and intensifies precipitation in downstream areas, cools the surface of the lower Bohai Sea and the border area between China and North Korea by Rosebery wave transport, and even impacts global climate (Liu et al., 2020; Xu et al., 2018; Xie et al., 2020; Liu et al., 2019b; Wang et al., 2020; Han et al., 2009; Sun et al., 2017).

However, the TP itself is an important source of dust, with severe desertification and many sand dunes in various regions (Fang, 2004; Li et al., 2001). Observations show that dust events on the plateau tend to occur in spring and winter, and that the dust bands tend to move northward (Han et al., 2008). Owing to the climatic sensitivity of the plateau and its influence on global climate, it is important to quantify the contribution of dust outside and within the TP to the atmosphere above the plateau. At present, there are a few studies on dust inner the plateau, and conclusions regarding the distribution of dust sources inner the plateau, the spatial and temporal variability of dust initiation, and the contribution of dust to the air column above the TP are still controversial. The harsh natural environment of the plateau limits ground-based observations, and satellite observations do not provide adequate information regarding the sand initiation process and the amount of sand initiation in the interior of the plateau owing to low resolution and instrumental observation techniques. The Weather Research and Forecasting model with Chemistry (WRF-Chem) is based on a mesoscale weather model and can partially compensate for the temporal and spatial deficiencies of the observations by simulating the meteorological field while reproducing the long-range transport of dust, providing quantitative information on dust emissions and radiative impact. In this study, the main body of the plateau was divided into three regions, the western, southern, and northern parts, based on surface and climate characteristics. A quantitative study of the spatial and temporal variability of dust fluxes and atmospheric dust properties of different climatic regions in the interior of the plateau was conducted



using WRF-Chem in three regions over a three-year period (2004-2006) with high resolution. Details regarding model construction, data processing and zoning are presented in the next section. Section 3 assesses the accuracy of the simulation results, and in Section 4 we analyze the temporal and spatial variability of sand fluxes, followed by a study of the annual variability and vertical distribution of dust column concentrations over the plateau. A summary of our findings is presented in Section 5.

## 2. Study region and data

### 2.1 Study region

The study area was divided to cover nearly the entire plateau region and exclude the influence of surrounding dust and sand source sites as much as possible. The widespread sand dunes and deserted land on the TP provide a large amount of raw material for the generation of dust storms (Fang, 2004; 105 Han et al., 2009). Because different regions of the plateau have different climates, Jiang (Jiang and Wang, 2001) analyzed climatic variations in surface temperature, precipitation, and monsoon on the plateau and pointed out that 85°E and 33°N are the dividing lines for spatial variations in surface temperature on the plateau; the surface temperature gradually decreases from west to east on the west side of 85°E, gradually increases from west to east on the east side of 85°E, gradually decreases from 110 north to south on the north side of 33°N, and gradually increases from north to south on the south side of 33°N. The spatial distribution of precipitation is consistent with surface temperature. In addition, 33°N is close to the arid and semi-arid regional divide on the TP under ecogeographic zoning, which also shows a clear regional divide in the plateau as per precipitation and vegetation cover types (Zheng, 1996; Xu et al., 2013). Therefore, we divided the study area into three regions according to climatic 115 characteristics (Figure 1): the western region (31.15-36.07°N; 79.74-85°E), northern region (33-40°N; 85-105°E), and southern region (27-33°N; 85-105°E). The three areas cover a large part of the main plateau body excluding as much of the surrounding sand sources as possible.

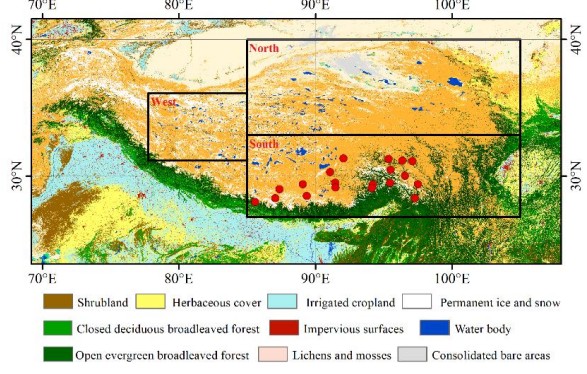

Figure 1 Division of the Tibetan Plateau study area. The red dots on the map are the geographical locations of the 19 selected sites.



## 2.2 Data

### 2.2.1 Model description

The WRF-Chem model (V3.9.1) is based on the WRF model with the introduction of the Chem module and full on-line coupling of meteorology and chemistry (Grell et al., 2005). In this study, the simulated area had a horizontal resolution of 6 km, with 99 × 99 grid points in the west, 99 × 334 grid points in the south and 115 × 334 grid points in the north. The depth from the surface to 50 hPa was vertically divided into 40. The simulation process began on January 1, 2004, with monthly updates of the initial field, and 24 h after 00:00 on the last day of the previous month was used to allow the model to spin up. The boundary and initial conditions were obtained from FNL global reanalysis data provided by the National Centers for Environmental Prediction at a resolution of 1°, which were available every 6 h. The model was output in netcdf format at half-hour intervals. The following parameters were chosen for the simulation: Purdue Lin microphysical scheme (Gustafson et al., 2007), Yonsei University planetary boundary layer scheme (Noh et al., 2006), RRTMG long-wave and short-wave radiation scheme (Mlawer et al., 1997; Iacono et al., 2000), Noah scheme (Chen et al., 1996), and Grell cumulus scheme (Grell et al., 1994); other parameter settings refer to Chen (Chen et al., 2013) and Zhao (Zhao et al., 2020).

GOCART directly calculates the sand initiation flux for each grain size based on empirical equations for the vertical dust flux and wind speed (Ginoux et al., 2001), and the dust emission flux equation is calculated as follows:

$$G = CSs_\mathrm{p}U_{10}^2 \ (U_{10}-U_\mathrm{t})$$

where $C$ ($\mu$gs$^2$m$^{-5}$) is a constant; $S$ is a source function that defines the area of potential dust sources, consisting of vegetation, snow, and other surface factors; $s_\mathrm{p}$ is the proportion of dust emissions for the different classifications. Dust was classified into five classes in the GOCART scheme, with particle size intervals of 0.2-2, 2-3.6, 3.6-6, 6-12 and 12-20 $\mu$m. $U_{10}$ is the horizontal wind speed at 10 m above ground level. $U_\mathrm{t}$ is the threshold wind speed; when the wind speed is less than this value, the dust will not be blown up into the atmosphere. The threshold wind speed is a function of particle size, air density, and surface soil moisture.

### 2.2.1 Observational data

The MERRA-2 reanalysis data were provided by NASA and the data started in 1980 with a horizontal resolution of 0.5° × 0.625° and 72 layers in the vertical direction. In this study, MERRA-2 monthly averaged data were selected to validate the model simulation results for temperature at 2 m above the ground ($T_2$) and wind speed at 10 m ($U_{10}$) to assess the accuracy of the simulation.

In addition, wind speed and temperature observations from 19 stations in the Tibetan region (Nagqu, Dangxiong, Lazi, Namling, Mozhukongka, Zedang, Nyalam, Tingri, Gyantse, Dingqing, Zuqi,



Changdu, Lolong, Bomi, Baju, Linzhi, Milin, Zogong, and Tsumu) provided by the China
Meteorological Data Network were used to evaluate the model results.

## 3. Model evaluation

The model outputs of $T_2$ and $U_{10}$ at the ground level were averaged and compared with the
monthly MERRA-2 reanalysis data. December to February, March to May, June to August and
September to November were classified as winter, spring, summer, and autumn, respectively. As the
160 spatial resolution of the reanalysis data is much lower than the model output, only an approximate
comparison of the characteristics of the variables in terms of spatial distribution was possible, and
there was no detailed comparison. Figure 2 shows that the simulation results were consistent with the
spatial distribution of the reanalysis data in terms of both seasonal variation and distribution of high-
value areas. Temperature was highest in summer and lowest in winter at 2 m above ground level in the
165 TP, with spring and autumn temperatures being within this range. $T_2$ gradually decreased from the
southeast to northwest of the TP. Both the reanalysis data and model simulation results showed that
the Qaidam Basin is a high-value area for $T_2$ on the plateau. In contrast to that seen for $T_2$, the highest
and lowest wind speeds occurred over the plateau in winter and summer, respectively, with the
northwestern part of the plateau being the high-value area.

The results of the three regional models were gridded and averaged to compare the annual
variation in the regional average characteristics with the corresponding areas in the reanalysis data
(Figure 3). The two datasets fit well for the annual variations in $T_2$ in the three regions, both showing
a single-peaked structure, with the highest temperatures in July, reaching 278.87, 284.45, and 285.49
K in the western, southern and northern regions, respectively. The lowest temperatures were recorded
in winter, with the three regions reaching 253.97, 265.89, and 259.95 K, respectively. The model results
were slightly lower than the reanalysis results. The annual variation in $U_{10}$ was opposite that of $T_2$,
with the maximum wind speed occurring in January, with maximum wind speeds of 4.9, 3.3, and
3.4 m/s in the three regions, and then decreasing month by month to a minimum in August, with
minimum wind speeds of 2.2, 1.9, and 2.3 m/s in the three regions. The model results and reanalysis
data showed a significant difference in wind speed in summer, with the model results being
significantly lower than the reanalysis data, with a difference of 1 m/s.

The China Meteorological Data Network provides monthly averages of multiyear observations of
meteorological elements at 19 stations in TP. $T_2$ and $U_{10}$ were also very close to the observed heights
of the daily variables. In this study, the observations were compared with the model results to increase
the credibility of the model results. However, because almost all stations were located in the
southeastern part of the plateau, only the southern region was selected for comparison. The results of





the comparison (Figure 4) showed that the annual trends of the three types of $T_2$ data in the southern region were very consistent; however, in the comparison of wind speeds, the model results were closer to the observations in summer, and the wind speeds were lower in the reanalysis, whereas in winter, the model results fit better with the reanalysis data, and the observations were significantly low. Because of the need to consider the suitability of the working environment when setting up the stations, it is possible that no stations were set up in the windy southern regions in winter, resulting in low winter wind speed observations. In general, the model results simulated the spatial distribution and annual variability of the meteorological elements very well, but because of the long duration of the simulation and the high spatial resolution required, the limitations of the observed data make it difficult to meet the above requirements, and the model data will be used to further investigate the spatial distribution and temporal variability of sand fluxes.

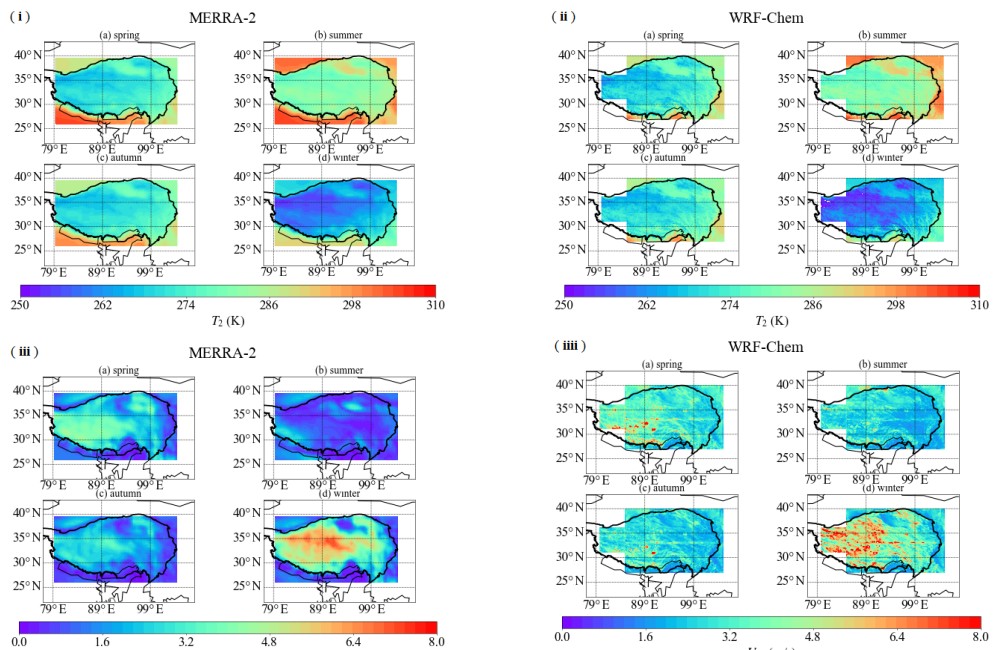

Figure 2 Comparisons of average spatial distribution of temperature at 2 m above the ground ($T_2$) between the MERRA-2 (i) and the WRF-Chem (ii), and comparisons of wind at 10 m above the ground ($U_{10}$) between the MERRA-2 (iii) and the WRF-Chem (iiii) in the three regions on the Tibetan Plateau





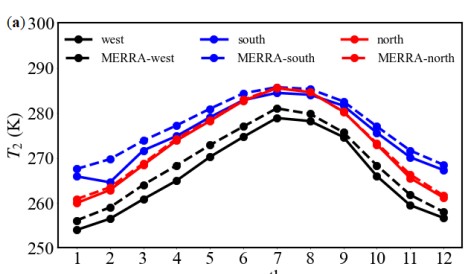

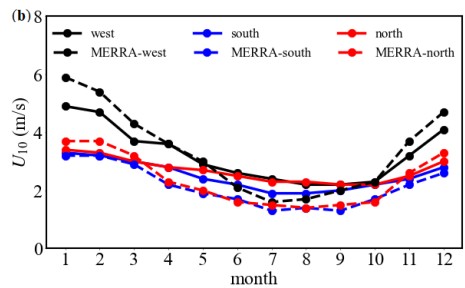

Figure 3 Comparison of mean annual changes in temperature at 2 m above the ground ($T_2$) (a) and wind at 10 m above the ground ($U_{10}$) (b) in the three regions on the Tibetan Plateau

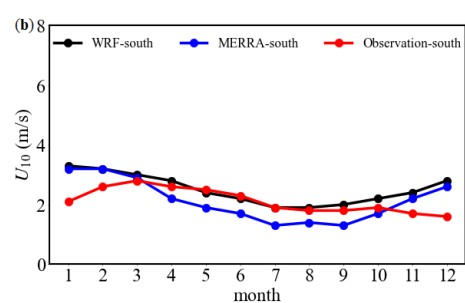

Figure 4 Comparison of mean annual changes in the temperature at 2 m above the ground ($T_2$) (a) and wind at 10 m above the ground ($U_{10}$) (b) in the three regions on the Tibetan Plateau

## 4. Results

### 4.1 Dust emission over the TP

The dust emissions during the four seasons in the different regions were calculated using the dust emission flux. The seasonal variation of the spatial distribution of dust emissions in the western region (Figure 5 (i)) shows that in winter, almost all the areas in the western region had dust emissions greater than $5.50 \times 10^8$ μg·m$^{-2}$, except for the area where the Gangdise Mountains are located, where the sand load is low because of the large difference in altitude; in summer, dust emissions decreased significantly, with dust emissions in most of the regions being approximately $3.50 \times 10^7$ μg·m$^{-2}$. Although there was an overall reduction in the amount of sand generated during the summer, dust emissions in some areas still reached values above $1.00 \times 10^8$ μg·m$^{-2}$, which is a high amount of dust emissions. These areas were located in the source areas of the Yarlung Tsangpo River and the northern part of the Qiangtang Plateau. This high-value area of year-round dust emissions can be considered as the source area of dust within the TP; these high-value areas occupy almost the entire western part and contribute up to 70% of the dust and sand over the plateau in February and March (Mao et al., 2012).

The spatial distribution of dust emissions in the southern region is very similar to the southern topography (Figure 5 (ii)), and the plateau profile can be seen more clearly in the south and east.



Similar to that in the west, the high values of dust emissions in the southern region are primarily in winter and can reach above $4.00 \times 10^8$ μg·m$^{-2}$ in the high-value area, whereas the dust emissions in summer are less, with emissions in most of the area being less than $3.50 \times 10^6$ μg·m$^{-2}$. In summer, the southern Tibetan valley floor and the Yarlung Tsangpo River basin, together with the Namucuo region to the north can be seen as high-value areas for dust emissions. In contrast, the Everest region and the connected cities of Shannan, Linzhi, and Sichuan had low dust emissions values throughout the year.

The spatial distribution of dust emissions in the northern region also clearly showed the northern outline of the TP, with dust emissions reaching more than $4.50 \times 10^8$ μg·m$^{-2}$ in winter and approximately $5.00 \times 10^7$ μg·m$^{-2}$ in summer (Figure 5 (iii)). In summer, the simulated high dust emission areas in the northern region were the Yangtze and Yellow River source basins, Qaidam Basin, and Qinghai Lake and its surrounding areas, which are consistent with the observed areas of frequent dust storms and the high value areas of horizontal dust transport fluxes simulated by the model (Du et al., 2022). However, the Qaidam Basin was not a high-value area for dust emissions in winter, probably because dust emissions are primarily affected by wind speed. The Qaidam Basin is less affected by cold air in winter because of the altitude difference and being surrounded by high mountains; therefore, there is less dust in the Qaidam Basin in winter. In spring, the Qaidam Basin is primarily affected by the strong cold air invasion from Xinjiang, the cold front transit, which causes greater chances of windy weather on the ground. Most areas of the basin are affected by low precipitation in spring, warming up, thawing soil, and bare ground, which will bring windy weather to the local area, and dust is be easily formed under the influence of windy weather. In summer, the Qaidam Basin is primarily affected by the East Asian summer wind, and local heavy rainfall, hail, and other strong convective weather are prone to occur, thus causing local windy weather (Zhang et al., 2014; Xu et al., 2011).

In general, the spatial distribution shows that the western part has the most dust the three regions and the southern part is the least dusty area.



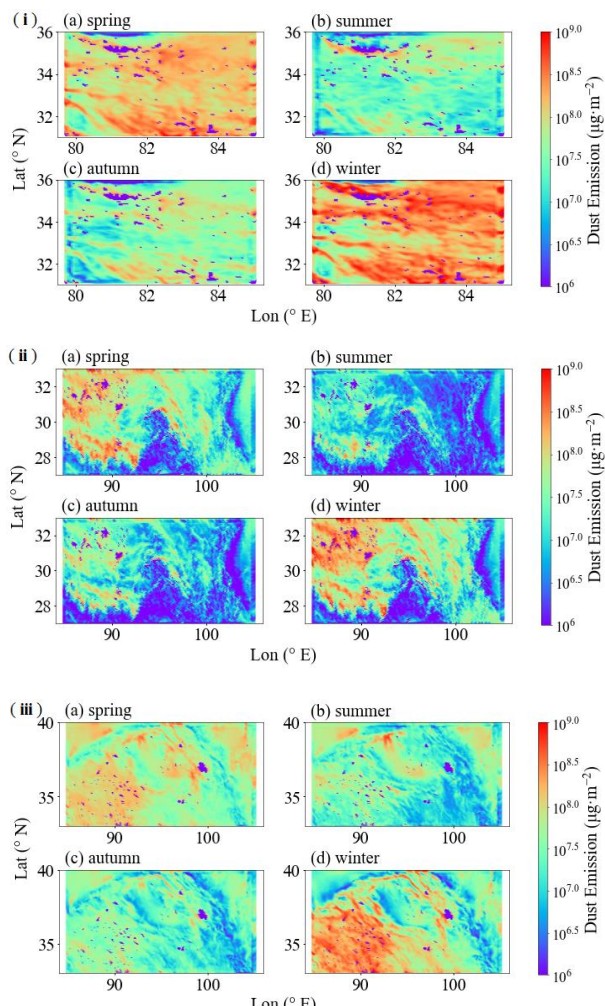

Figure 5 Seasonal variations in the spatial distribution of dust emissions over western region (i), southern region (ii), and

northern region(iii) of the Tibetan Plateau

## 4.2 Annual variation and differences in dust emissions and loading

Wind speed is an important factor influencing the dust emission process, and the annual variations in dust emissions in the three regions of the TP (Figure 6) were more similar to the annual variations in wind speed at 10 m above ground; the maximum dust emission in January in the three regions reached $11.00 \times 10^7$, $3.30 \times 10^7$, and $4.50 \times 10^7$ $\mu g \cdot m^{-2}$ respectively, then decreases month by month.

In the west, the minimum value of $0.82 \times 10^7$ $\mu g \cdot m^{-2}$ was reached in August, in the south, the minimum value was $0.27 \times 10^7$ $\mu g \cdot m^{-2}$ in July, and in the north, the minimum value was $0.85 \times 10^7$ $\mu g \cdot m^{-2}$ in October. This annual trend was consistent with the observed annual trend in dust storm frequency (Han et al., 2008; Du et al., 2022). Among the three regions, dust emission are greatest in the west, especially



in winter, when they were twice as high as in the other two regions.

However, there were significant differences in the annual variations in dust loading in the three regions (Figure 7), with dust loading being the accumulated dust content of the entire atmosphere over a season. In terms of trends, the maximum dust load in the west occurred in January at $94.00 \times 10^5$ µg·m$^{-2}$ and then decreased rapidly to below $10.00 \times 10^5$ µg·m$^{-2}$, with a small peak in summer and continues growth through November. In the south, there was no clear annual trend but fluctuations throughout the year, and with values as small as 36.00 µg·m$^{-2}$, it can be assumed that there was almost no dust in the atmosphere. In the north, the dust load is much larger than in the west and south, and the annual trend was the opposite of that in the west, with low values in winter but also reaching $0.44 \times 10^{10}$ µg·m$^{-2}$. There were two peaks of up to $13.00 \times 10^{10}$ µg·m$^{-2}$ in April and August. Wu (Wu et al., 2019) used multiple models to simulate the annual variation in the multiyear average optical thickness of dust over the plateau; they also showed a similar bimodal structure.

Although the annual trends of dust emissions were similar, the amount of dust that remained in the atmosphere showed significant differences, not only in the annual trends. In the west and south, the dust load was less than the amount of dust emissions, especially in the south. While in the northern region the dust load was much higher than the amount of dust emissions, indicating that the dust over the western and southern parts of the plateau stayed in the atmosphere for a short period of time, with the dust being blown up into the atmosphere and then rapidly falling back to the plateau surface; the dust in the north stayed in the atmosphere for a long period of time.

We used the amounts of dust emissions and loading to obtain a difference ($C$) with the soil moisture and $T_2$ and $U_{10}$ to analyze the factors that may affect dust deposition and dispersion. The results (Table 1) show that in each region, soil moisture ($W$) and $T_2$ are negatively correlated with $C$ indicating that the higher the soil moisture and $T_2$, the more readily dust stays in the air. Janae (Csavina et al., 2014) found that dust concentration in the atmosphere increased with an increase in relative humidity under high wind speed conditions, and many articles on the influence of meteorological elements on dust concentrations have attributed the effect of relative humidity on dust concentrations to the fact that relative humidity affects soil moisture and thus reduces the ability of dust to stick (Kim and Choi, 2015; Ju et al., 2018). Temperature affects the mixing layer height, when the temperature rises the mixing layer height also increases, at the same time the uplift of the plateau enhances the transport of sand and dust to higher altitudes, especially with the summer heat effect, allowing more of the dust to reach the upper troposphere and be transmitted. In addition, higher ground temperatures increase the relative humidity of the air as soil moisture evaporates, making it easier for the dust to remain in the atmosphere (Hussein et al., 2006; Yang et al., 2019). Furthermore $U_{10}$ was positively correlated with $C$, indicating that the higher the $U_{10}$ value, the less likely that dust remained in the air



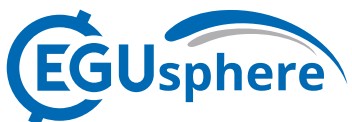

and that high winds would rapidly blow the dust away from the source. Of the three regions, the western region had the least correlation and the northern region had the highest correlation. This may be related

to the complexity of the natural environment in different regions, where there are many sand dunes and permafrost in the winter, and the fact that the western landscape is hilly, which may lead to obstruction of dust transport. The southern region is wet year-round, with lush ground vegetation and snow-covered mountains; therefore, there are more factors influencing dust deposition and dispersion and the correlation is low. However, the correlation between $T_2$ and $C$ was high in all three regions,

reaching 0.91 in the north, indicating that it is one of the main factors influencing dust deposition and dispersion.

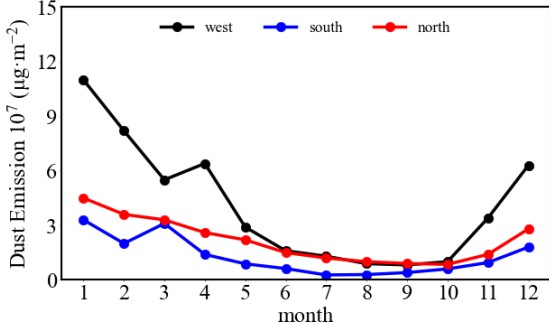

Figure 6 Annual variation in dust emissions in three regions on the Tibetan Plateau

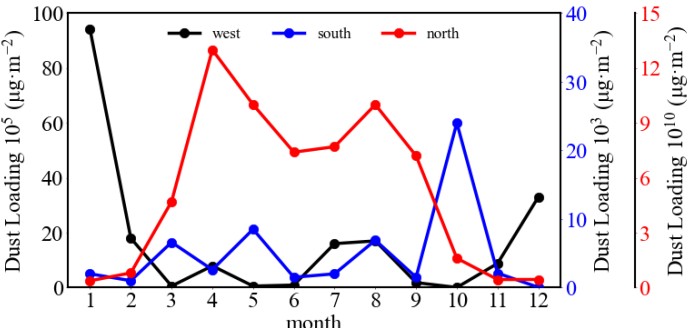

Figure 7 Annual variation in dust loading in three regions on the Tibetan Plateau



Table 1 *C of the three regions correlates with soil moisture, $T_2$, and $U_{10}$*

|  | $W$ | $T_2$ | $U_{10}$ |
|---|---|---|---|
| West | -0.06 | -0.34 | 0.13 |
| South | -0.41 | -0.51 | 0.57 |
| North | -0.80 | -0.91 | 0.70 |

### 4.3 Dust transmission over the TP

The height of the top of the boundary layer over the TP has been given as approximately 8 km above sea level (Chen et al., 2013; Zhang et al., 2020a). Some studies have also found that the vertical wind speed profile at 300 hPa above the plateau is almost always an updraft, and westerly winds are prevalent, making the dust reaching this height less likely to settle back to the plateau surface and more conducive to outward transport (Xu et al., 2018; Ji et al., 2015). Xu (Xu et al., 2018) pointed out that the optical thickness and dust mass fluxes of dust below 3 km and above 8 km of the atmosphere are very different, with dust above 8 km spreading downstream of the plateau. Therefore, we chose 8 km as the cutoff line and suggested that when the dust rises to 8 km, it will no longer settle back to the plateau surface but will remain in the atmospheric nodule or be transported downstream with circulation.

Figure 8 shows that summer was the period with the highest percentage of dust above 8 km in the three regions, with the lowest percentage of dust in winter. Almost no dust was present above 8 km in winter in the south and north, and 2.19% above 8 km in the west. In the south, the percentage can reach 42% in summer; however, with the already low dust concentrations in the south throughout the year, that atmosphere can be considered almost dust free. This trend does not reflect the problem well; therefore, the analysis will be done primarily for the west and north. In the north, dust loading reached $2.00 \times 10^6$ µg·m⁻² in winter, and the proportion of dust above 8 km remained at 0%, indicating that dust accumulates near the surface in winter on the plateau, is difficult to transport over long distances, and remain, in the atmosphere for long periods. The proportion of dust in both regions started to increase after March in spring and remained high, approximately 10%, in summer, with a relatively consistent annual trend. The vertical profiles of boundary layer height, dust concentration, and wind at 33.5°N in summer (Figure 9) showed that in summer, the top height of the boundary layer in the western part of the plateau was approximately 5.5 km and that dust concentrations reached 289.00 µg·m⁻³ at the surface west of 82°E. The dust concentration decreased with increasing height, subsidence predominated and the horizontal wind was predominantly westerly, allowing dust to be transported downstream. Significant updrafts could be seen east of 82°E, with dust being lifted off the ground and transported upward, and high dust concentrations occurring near the top of the boundary layer and continuing upward through the boundary layer, with dust concentrations reaching up to 234.00 µg·m⁻³. Many





studies have found that a distinct aerosol belt exists near the top of the troposphere during the summer monsoon in the plateau, and that the heat pumping effect of the plateau in summer, combined with the stable presence of anticyclones over the plateau during the monsoon, makes it easier for dust

transported to the upper layers to travel outwards, especially downstream (Park et al., 2009; Xu et al., 2018; Zhang et al., 2020b). In conclusion, summer is the time when dust is most actively transported outward in the upper troposphere of the TP; however, this only accounts for 10% of the dust aerosols in the interior of the TP.

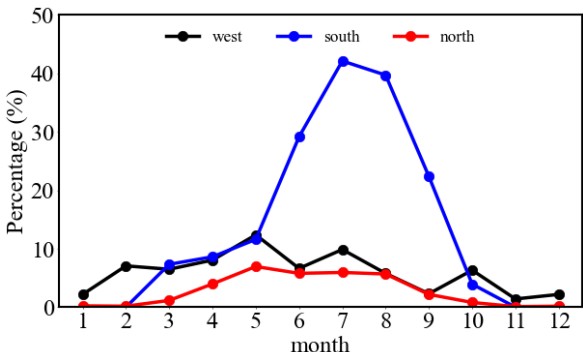

Figure 8 Percentage of sand and dust above 8km in three regions on the Tibetan Plateau

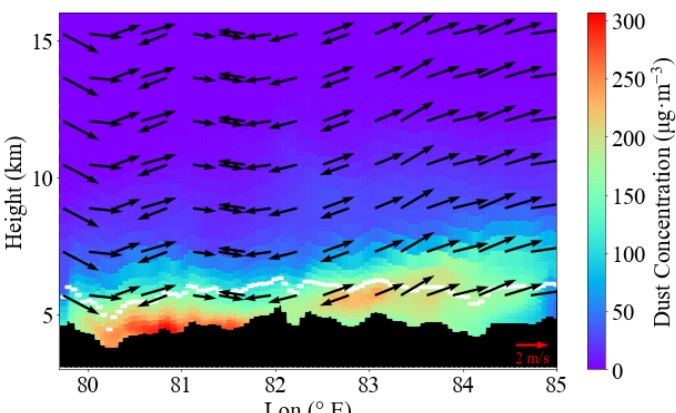

Figure 9 Vertical profiles of boundary layer height, dust concentration and wind at 33.5°N in summer in the western region of the

Tibetan Plateau, where the white dashed line is the top boundary layer height and the wind profile is the effect of expanding the

vertical wind speed by a factor of 10

## 5. Conclusions

In this study, the temperature and precipitation characteristics of the Tibetan Plateau were used to

divide the plateau into three regions, namely the western, southern, and northern regions, and the WRF-Chem model was used to study and analyze dust emissions, dust loading, and their characteristics in the three different climatic regions from 2004 to 2006. The results of the simulations were generally





consistent with the reanalysis data and ground observations in terms of the spatial distribution and annual variation of $T_2$ and $U_{10}$, reflecting the variation characteristics of high wind speeds in winter, low wind speeds in summer, and low temperatures in winter, high temperatures in summer. The conclusions are as follows:

1.  The spatial distribution of dust emissions shows that winter is a period of widespread dust onset in the three regions, and dust emissions decreased significantly in summer. Higher dust emissions can still be seen in the dust source areas during the weakening summer period, which are located in the northern part of the Qiangtang Plateau in the western part of the plateau, the Yarlung Tsangpo River basin, the Namucuo and Lhasa regions, Qaidam Basin, the source areas of the Yellow and Yangtze Rivers, and the Qinghai Lake and its surrounding areas. The dust-generating areas were primarily concentrated in the west and north, with fewer dust-generating areas in the south.

2.  The annual variation in dust emissions was more consistent with the annual trend in $U_{10}$, with the highest dust emissions starting in January, reaching $11.00 \times 10^7$, $3.30 \times 10^7$, and $4.50 \times 10^7$ µg·m$^{-2}$ in the west, south, and north, respectively, with different values for the lower months of $0.82 \times 10^7$, $0.27 \times 10^7$, and $0.85 \times 10^7$ µg·m$^{-2}$. Dust emissions were the highest in the west and lowest in the south. The annual trend of dust loading varied significantly, with the peak in the west occurring in January and remaining at a low value thereafter. The dust loading in the south fluctuated throughout the year with no evident trend, and the north showed a bimodal structure with peaks occurring in April and October. There was also a significant difference in the magnitude of the dust load in the three regions, with the southern region having a low level of dust in the atmosphere, the northern region reaching $13.00 \times 10^{10}$ µg·m$^{-2}$, and the west in between. The factors influencing the difference between sand initiation and dust load are complex but are primarily influenced by the temperature at 2 m above ground level.

3.  The peak proportion of dust above 8 km in the atmosphere in the three regions occurred in summer, with the proportion of dust above 8 km in the atmosphere in summer being 12% and 7% in the western and northern regions, respectively, and reaching 42% in the south. In winter, the dust content in the upper layers of the plateau atmosphere was low, with only the western part accounting for 2.2%. In summer, the upper troposphere of the plateau is active in dust transport, with strong upward movement over the plateau; thus, dust can penetrate the boundary layer and be transported upward strongly, together with outward transport by anticyclones over the plateau.



**Author Contributions**

YSZ, JNL, and LZ designed research; YSZ, ZDZ, BEL, HTZ., PFT, and XJC performed research;
YSZ, JNL wrote the paper

**Competing Interest Statement**

The contact author has declared that none of the authors has any competing interests.

**Acknowledgements**

We thank to the MERRA-2 reanalysis data were provided by NASA (https://disc.gsfc.nasa.gov) and
the wind speed and temperature observations from 19 stations in the Tibetan region provided by the
China Meteorological Data Network (http://data.cma.cn/). We also thank the Land cover products
(https://zenodo.org/record/4280923#.Yzg4dmu-vqA). The authors acknowledge support from the
Second Tibetan Plateau Scientific Expedition and Research Program (STEP) (No. 2019QZKK0602)
and support from Supercomputing Center of Lanzhou University. We also acknowledge the editor
Stelios Kazadzis and two referees.

**Financial support**

This research has been supported by the Second Tibetan Plateau Scientific Expedition and Research
Program (STEP) (No. 2019QZKK0602).

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
