# Peer review of "The role of dust sources in the Tibetan Plateau may be Underestimated: A high-resolution simulation study of regional differences in dust characteristics in the Tibetan Plateau"

_EGUsphere, 2022_

## Referee Comment (RC2)

**Introduction**

**1. The introduction part is missing the literature survey on dust simulations using WRF model. The authors should provide a detail on how WRF model has been used for dust simulation in the past (please provide a brief description on the state-of-the-art on the use of WRF model for dust simulation (in Tibetan plateau or elsewhere)) and what new has been done in this manuscript.**

**2: Line 65: "from the northern part of the plateau to the plateau". What does the authors mean here? Please explain.**

**3: Line 82: "there are a few studies on dust inner the plateau". May be this sentence needs restructuring.**

**Study region and Data**

**4: Line 114: "Therefore, we divided the study area into three regions according to climatic characteristics (Figure 1)". The previous lines have references Zheng, 1996; Xu et al., 2013. Which one of these does the authors have used for "climatic characteristic". If some other reference is used, then please mention here.**

**5: The methodology used to perform the WRF simulations has not been explained well in this manuscript. The authors provide a brief description of the schemes that they have used in Section 2.2.1 But they have not provided any details on the on-line coupling of the WRF-Chem model as they mention in line 123. The authors are suggested to provide a detail on the simulations performed and how the on-line coupling of meteorology and chemistry is performed. Also, the authors should provide details on how the simulations are performed for the three different regions mentioned in Section 2.1. Are they performed independent of each other or through nested domains?? If the simulations are performed independently for each domain, then the authors should provide a reasoning for this i.e., if the boundary conditions will be accurately represented for each domain. If nested domains are used, then the authors should provide details of these simulations.**

**Model evaluation:**

**6: Line 170: "three regional models". What does the author mean by "three regional models"?? The authors are only using WRF-Chem model in this manuscript. May be the authors are referring**

to the three regions of the model. If not, then provide an explanation here. Please rephrase this sentence to avoid confusion.

**6: Line 175: "The model results were slightly lower than the reanalysis results.". This means that the model is underpredicting these values as comapred to the reanalysis values. Can the authors provide a probable reason for this underprediction??**

**8: Line 179: "The model results and reanalysis data showed a significant difference in wind speed in summer, with the model results being significantly lower than the reanalysis data, with a difference of 1 m/s". Can the authors provide an explanation for this finding?**

**9: Figure 3: There is something wrong with the legend of this figure. What does the dashed line and the solid lines represent?? The legend only has description of solid lines.**

**Results:**

**10: Line 248: "maximum dust emission in January". Can authors explain the reason why the dust emission was maximum in a winter month of January and it was lower in summer months, in fact minimum in summer month of August. Why is the emission maximum in Winter and not in summer?? If the frequency of dust storms are more in winter months, then is it possible that the maximum dust emission values are arising due to the frequency, but is it possible that the dust emissions are more in summers but when averaged for the month, then the values comes out to be less?? If so, then has the authors take the frequency of these events into account?? Also, these emission are at what level (at the surface or the averaged values over the 40 vertical layers used in the simulation)??**

In line 252, the authors mention "This annual trend was consistent with the observed annual trend in dust storm frequency". But has the frequency of occurrence of these events taken into account and considered while averaging the emissions for a month?

**11: Line 263: "Wu (Wu et al., 2019) used multiple models to simulate the annual variation in the multiyear average optical thickness of dust over the plateau; they also showed a similar bimodal structure.". In this work by Wu et. al., authors have presented that for TP, the dust emission is more in March-April than in January. While in this (current manuscript) work, the authors have presented**

that the emissions are maximum in January and then decreases to a minimum in August. Can the authors support this claim??? Also, which bimodal structure are the authors talking about??? Please explain this.

**12: Figure 7: I assume that the solid lines represent the model simulation in figure 3, then in North region the simulated temperature is higher from April to September, while wind speed remains mostly constant. Figure 6 shows that the dust emission is low in these months. Can the authors explain why the dust loading in North region is quite high from April to September (Figure 7).**
**13: Figure 8: How is the explanation provided for Figure 8 related to Figure 7? It would be nice if the authors can link these two figure alongwith a proper explanation.**

---

## Author Comment (AC1)

This manuscript reports results of WRF model simulation to quantify the spatial and temporal variability of dust emissions, dust loading, and their characteristics within the Tibetan Plateau during 2004-2006. The authors divide the main body of the Tibetan Plateau into three regions, and argue that the results of the simulations about dust activity were consistent with the reanalysis data and ground observations.

Overall, simulation of winter dust on the Tibetan Plateau is with great importance. However, this manuscript has some shortages, and some of the implications and discussions in the manuscript are not appropriate. I have a number of comments and concerns listed below, and the authors should make Major revisions of this manuscript before it can be potential published on ACPD.

*[Response]* Thank you very much for your suggestion. In particular, many references are provided that allows me to verify the research results and modify next steps of my research plan. We have taken all of the specific comments into account in the revised version of the manuscript. Please see the detailed response marked blue below.

**Main comments**

1.The current definition of the Tibetan Plateau is confusing and not appropriate. In common sense, the Tibetan Plateau is the region higher than 2500 or 3000m above sea level. The western of the Sichuan Basin, the western of the Loess Plateau, the eastern of the Tarim Basin and other regions shown in Fig. 1 cannot be incorporate into the Tibetan Plateau. Please read the definition by Yao et al., 2012, Nature Climate Change (DOI: 10.1038/NCLIMATE1580) and Liu et al., 2022, Global andPlanetary Change (https://doi.org/10.1016/j.gloplacha.2022.103893).

*[Response]* Thank you for your comment. In this manuscript, which definition of the Tibetan Plateau we chose is that the region extends from the Pamir Plateau in the west to the Hengduan Mountains in the east, from the southern edge of the Himalayas in the south to the northern side of the Kunlun-Qilian Mountains in the north. Li Bingyuan (1987) discussed the principles and specific boundaries for determining the extent of the Tibetan Plateau in a more systematic manner, and proposed that the basic principles for determining the extent of the plateau should be based on the geomorphological features, the plateau surface and its altitude, while taking into account the integrity of the mountains. Zhang Yili

(2002) argued for the principles of determining the extent and boundaries of the Tibetan Plateau based on new research results in related fields and years of field practice, and combined with information technology methods to precisely locate and quantitatively analyze the extent and location of the boundaries of the Tibetan Plateau, then they put forward this definition of the Tibetan Plateau. ( http://data.tpdc.ac.cn/zh-hans/data/61701a2b-31e5-41bf-b0a3-607c2a9bd3b3/)

Actually, the region higher than 2500 or 3000m above sea level was widely used by many scholars, but we chose the definition mentioned above because we want to cover the main body of the Tibetan Plateau as much as possible, meanwhile avoiding the influence of large drop in elevation. Besides, WRF-Chem can only provide regular rectangular areas for simulation, grid point with a resolution of 6 km is the compromise between resolution and simulated regions so that there are much more other regions were considered in simulation. Therefore, the difference of area between two definitions on this region could be ignored compared with the giant body of the Tibetan Plateau.

2.The division of the three regions on the Tibetan Plateau is crude. The authors mentioned that the resolution of WRF model was 6 km, it can be more precise for division of those three regions. The dust sources of the plateau are located in the northern part of the Qiangtang Plateau, the Yarlung Tsangpo River basin, the Namucuo and Lhasa regions, the Qaidam Basin, the source areas of the Yellow and Yangtze Rivers, and the Qinghai Lake and its surrounding areas. Therefore, Istrongly suggest the authors use Qiangtang Plateau (between the Kunlun Mountains and Gangdisi Mountains) as the western Tibetan Plateau, north of the Tanggula Mountains (roughly around 32-33 °N) as the Northern Tibetan Plateau, and south of the Gangdisi Mountains-Tanggula Mountains as the Southern Tibetan Plateau, on the basis of the definition of the Tibetan Plateau (higher than 3000m or 2500 m).

*[Response]* Thank you for your comment. There has been controversy over the division of the three regions on the Tibetan Plateau. The original intention in designing the experiment is that we want to ensure the remote inland in the western Tibetan Plateau is a source of sand and dust (such as Ali and its surrounding areas) and make a quantitative description of dust emissions in this area. Our team have carried out long-term aerosol investigation and observation experiments in the Ali region (Zhang et al., 2021, https://doi.org/10.1029/2020JD033286), during this time we found that Small dust events occur frequently in Ali and its surrounding areas. So when we divided regions, we tried to cover these areas and exclude surrounding deserts. As for the northern region and southern region we chose the 85 °E and 33 °N as the boundary because of the difference of temperature and precipitation. (Jiang and Wang,

2001, Jia et al., 2015).

But combined with the results of this research, the division of the three regions on the Tibetan Plateau from you could be as a good reference for the delineation of future research work areas. Thank you again.

3. The authors only provide results of three years during 2004-2006. However, a climate pattern (30 years, such as 1991-2020) can provide more confident results and reliable conclusions.

*[Response]* Wang and Zhang (2006) analysed the number of days occurred with dust and sandstorm weather (including floating dust, sand, dust storms and strong dust storms) based on observations from meteorological stations. The results show that there are two main areas of high dust and sandstorm weather occurrence in Northeast Asia during spring 2006, one occurring in the Taklamakan Desert and surrounding areas in southern Xingjiang, China, and the other being probed in the arid region of southeastern                                                                                              Mongolia, (https://kns.cnki.net/kcms/detail/detail.aspx?dbcode=CPFD&dbname=CPFD9908&filename=ZGQX 200610001000&uniplatform=NZKPT&v=AmhKSmhN9Hr4mkIG2ftYQjIhYCFvE0I3ZZDE2M8SR 2XxbFxuQzrQEE90QS_Owhk16TIlsFxDV78%3d). And a dust storm was observed on 22, Jan, 2006. So we simulate the Tibetan Plateau from 2004 to 2006 to quantify the spatial and temporal variability of dust within the plateau.

Obviously, a long duration simulation is important, but short simulations are also indispensable. Results of long simulations are prone to bias, Short-time simulations do not reflect trends. The results of this short simulations proved that this way to simulate dust emissions of the Tibetan Plateau is credible. So Long-term simulation about the Tibetan Plateau dust emissions will be my next work plan.

4. The dust emission rate data on the Tibetan Plateau throughout the manuscript should be checked carefully. Previous studies reported that dust emission rate in the Taklimakan desert was about 0.38 ton/ha yr (equals to 38 $g/m^2$), in the Central gobi-desert is about 0.24 ton/ha yr (equals to 24 $g/m^2$) (see Xuan et al., 2002, Atmospheric Environment, https://doi.org/10.1016/S1352-2310(02)00585-X). In comparison, the authors report dust emission rates of about $11.00 \times 10^7$ $\mu g \cdot m^{-2}$ (equals to 110 $g/m^2$) in the west, $3.30 \times 10^7$ $\mu g \cdot m^{-2}$ (equals to 33 $g/m^2$) in the south and $4.5 \times 10^7$ (equals to 45 $g/m^2$) in the

north during winter on the Tibetan Plateau, in this work. I am not sure whether dust emission rate on the Tibetan Plateau can be much higher than the Taklimakan desert and Go-bi desert?

*[Response]* Thank you for the reference that you recommended, I have carefully investigated this article and discussed the details with my co-authors. In this article, the authors use 30-year (1951–1980) climatological data and the modified US EPA empirical formulas to finish the research. The result that dust emission rate in the Taklimakan desert was about 0.38 ton/ha yr (equals to 38 $g/m^2$), in the Central gobi-desert is about 0.24 ton/ha yr (equals to 24 g/m2) is averaged over many years, it may have smoothed out some great values. Besides, the data between the reference and us used to simulate spanned a large time, from 1980 to 2004. And different models have various schemes of dynamic and thermal mechanism. So it is normal to see different results.

According to the latest research that the observations and simulated results showed that the QTP had a high dust emission rate, and the dust (PM10) emission intensity in the QTP during the period of 2000 to 2020 ranged from 0 to 1042 $g/m^2/y$ with a mean value of 32.4 $g/m^2/y$. (Du et al., 2022, https://www.sciencedirect.com/science/article/pii/S0016706122002373). And the averaged value from our simulations is 910 $g/m^2/y$, So I think the result is credible.

5. In the abstract, "few studies have been conducted on dust aerosols within the plateau", and Line 80-83, "there are few studies on dust within the plateau, and conclusions regarding the distribution of dust sources within the plateau, the spatial and temporal variability of dust initiation, and the contribution of dust to the air column above the TP are not clear". As far as I know, there are many studies on dust aerosols in the TP. The sentences are too absolute. For example, Liu et al., ACP 2008; Kang et al., AE 2016; Mao et al., SCIENCE CHINA Earth Sciences 2013;Mao et al., AE 2019; Yuan et al., JC 2019, etc. Mao et al., did a lot of work on dust using WRF-Chem in the TP. What are the differences between your manuscript and Mao's work? The authors should highlight your characteristics and differences relative to previous studies. In addition, please add the dynamic mechanism related to your results in the abstract.

*[Response]* Thank you for the references that you listed, I have investigated these articles carefully, and the text has been revised accordingly. Obviously, Mao's work has led to a number of important results and has provided much assistance in the design of the research work and validation of the results in this paper, this article also cites many of the results of his work (Mao et al., 2013, Feng et al., 2020).

But they analyzed the Tibetan Plateau as a whole in their simulations and chose a coarser grid resolution.

As stated in the introduction to our article, the complex topography and sub-surface conditions make the climate very different in different regions of the TP. So the characters of dust emissions and dust loading in sub-regions of the TP are significantly different. However, setting and analyzing the Tibetan Plateau as unity in simulation only reflects northern character only, because of the high value of dust loading of northern region. There are two pictures below, left one is dust optical depth (Wu et al., 2019, https://agupubs.onlinelibrary.wiley.com/doi/full/10.1029/2019JD030799), right one is dust loading of this article, they show that both the trend of dust optical depth in the Tibetan plateau and the trend of dust loading in the northern region are bimodal structure, but the trend of dust loading in the western and southern region are not. So sub-regional study of the Tibetan Plateau could be useful for future studies on the climatic effects of the Tibetan Plateau aerosols.

What's more, finer resolution can better reflect the details of the spatial distribution of dust on the Tibetan Plateau. And the abstract and introduction were modified.

[Figure]

6. If the authors add the model evaluation of AOD between the results of WRF and MODIS or MISR, cross sections of dust extinction coefficient between the results of WRF and CALIPSO, the results will be more reliable.

*[Response]* Thank you for your comment. According to annual variation in dust loading in three regions on the Tibetan Plateau (Figure 7), we compared data of average monthly AOD between the results of WRF and MODIS on April as following. They show that the simulation results were consistent with the spatial distribution of the MODIS data,but numerically lower than MODIS.

However, this article focused on the dust in the interior of the Tibetan Plateau while MODIS observe dust from global sources. Therefore, although comparison of the results verifies that the simulation data is reliable, this part of the result is not added to the article. If necessary, we will upload it to supplement.

[Figure]

7. The language of this manuscript needs editing by a native speaker.

[Response] The language of this manuscript was edited by a native speaker.

**Other comments**

1. Line 36, the definition of the Tibetan Plateau

[Response] The definition of the Tibetan Plateau has been introduction at main comment 1. And "The Tibetan Plateau (TP) is the highest-elevated landmark in the world, with elevations ranging from approximately 1500 to 5000 m above sea level." Can be found in Liu et al., 2019.

2. Line 45, Batangilin Desert? Is it Badanjilin Desert/ Badain Jaran Desert?

[Response] This error has been corrected

3. Line 53, the unit of 6.6G is "g" or "ton"?

[Response] This error has been corrected

4. Lines 79-82, the inner dust sources on the Tibetan Plateau are clear, please read Wu et al., 2013, Quaternary Science Reviews (http://dx.doi.org/10.1016/j.quascirev.2012.10.003).

[Response] Thank you for the references that you listed, I have studied these article carefully, this article present the detailed dust history from 1850 to 2004 AD on an annual timescale from a shallow

ice core from Tanggula, central Tibetan Plateau. The analysis leads to the conclusion that Tarim Basin and over the western Tibetan Plateau, are possible source areas. They also ensure that dust come from Udaoliang made up the largest proportion of the ice core record period. Tanggula Mountains are so far and high-elevated to be the representative of inland of the Tibetan Plateau, and the reestablishment of dust spatiotemporal distribution could be promoted with corporation between numerical simulation and climatological records, such as ice core. However, the dust sources and relative importance are still controversial, there are a lot of dust that cannot be transported to the Tanggula Mountains.

5. Line 93, similar to former major concern, why only choose 2004-2006?

*[Response]* The definition of the Tibetan Plateau has been introduction at main comment 3.

6. Lines 101-102, I strongly suggest the authors provide a map of sand dunes and deserted land on the Tibetan Plateau in the manuscript or in the supporting information

*[Response]* The map of sand dunes and deserted land on the Tibetan Plateau can be found in Fang, 2004; Han et al., 2009.

[Figure]

Fig. 1. Distribution of the modern and last glacial sand dunes, desertified sandy land, and the mean annual dust storm days and precipitation from 1961 to 2000 on the TP (black solid line: precipitation; black dashed: dust storm days; black circle including the number representing the following locations: 1, Guliya ice core; 2, Malan ice core; 3, Dunde ice core; 4, Delingha tree ring).

[Figure]

Fig. 1. The distribution map of modern and last glacial sand dunes, desertified sandy land and loess and the spatial mean annual days plot of dust storm from 1961 to 2000 on the Tibetan Plateau. Aeolian sand and loess distribution was drawn according to our field investigation and refs. [1~5], [9~12] and [19~21]. Dust storm data are from the Climate Center of China Meteorological Administration.

7. Lines 102-115, the climate patterns are the basis for the division of the three regions on the Tibetan Plateau. The authors should read some most recent papers about the climate modes on the Tibetan Plateau, redefine the three regions and rewrite this paragraph.

*[Response]* The definition of the Tibetan Plateau has been introduction at main comment 2.

8. Fig 1. The north and south are wrongly labeled, the authors should better add the variation of meteorological parameters such as temperature and precipitation of these three regions.

*[Response]* This error has been corrected, the variation of meteorological parameters such as temperature and precipitation of these three regions can be found in Jiang and Wang, 2001.

9. Line 206, Gondola Mountains?

*[Response]* This error has been corrected

10. Lines 236-236, are you sure that the modern East Asian summer monsoon can reach the Qaidam Basin and cause heavy rainfall? If yes, please cite the suitable references. To our knowledge, the maximum of the East Asian summer monsoon is around Lenglongling in the eastern Qilian Mountains.

*[Response]* The suitable reference was cited. Xv et al., 2011 (https://agupubs.onlinelibrary.wiley.com/doi/full/10.1029/2010JD015053)

11. Lines 271, "the dust in the north stayed in the atmosphere for a long period of time", does this mean the potential effects of dust from the Tarim basin?

*[Response]* In fact, some of the dust over the northern region indeed came from the Tarim Basin. But the Figure 5 in this article shows that there are many dust sources in the northern region. Compared with western region, more man-made events are reported here. And compared with southern region, precipitation is less here. These reasons make the dust in the north stayed in the atmosphere for a long period of time

12. Lines 289-291, the western Tibetan Plateau is mainly the Qiangtang Basin, the landscape consists planation surface

*[Response]* Our team has carried out long-term aerosol sampling and observation experiments in the Ali region (Zhang et al., 2021, https://doi.org/10.1029/2020JD033286). At present, our team are still sampling around the Tibetan Plateau. They did observe sand dunes and permafrost.

[Figure]

[Figure]

13. Lines 304-305, the height 8 km above sea level reaches the middle-upper troposphere, does this meant the height of the boundary layer over the Tibetan Plateau was about 3-4 km above the ground level?

*[Response]* The height of 8 km is come from Chen et al. 2013. We chose the highest altitude to be the dividing line in order to ensure dust that once overpasses it can't drop back to the ground.

[Figure]

**Figure 9.** (a–d) Vertical-latitude cross-section at 85.5°E of daily averaged aerosol extinction coefficient and potential temperature on 26–29 July from the WRF-Chem simulations over domain 1. Black contours indicate the potential temperature. The red line represents the daily averaged planetary boundary layer height (PBLH).

14. Please add the linkages of the meteorological and other dataset used in this paper in the Acknowledgement section.

*[Response]* The linkages of the meteorological and other dataset used in this paper in the Acknowledgement section were added.

15. References should be updated. For instance, line 489 "Journal of geophysical research. Atmospheres: JGR, 124, 8043-8064" is not correct.

*[Response]* This error has been corrected

16. Please check all the reference format. Like in line 55, 61, "the results of Wang(Wang et al., 2021)", "Hu(Hu et al., 2020)", not correct.

*[Response]* This error has been corrected

---

## Author Comment (AC2)

Thank you very much for your kind and helpful suggestion. We have taken all of the specific comments into account in the revised version of the manuscript. These suggestions make the manuscript more complete, logical and standard to publish on the scientific journal. Please see the detailed response marked blue below and the changes marked red in the revised manuscript.

**Introduction**

**1. The introduction part is missing the literature survey on dust simulations using WRF model. The authors should provide a detail on how WRF model has been used for dust simulation in the past (please provide a brief description on the state-of-the-art on the use of WRF model for dust simulation (in Tibetan plateau or elsewhere)) and what new has been done in this manuscript.**

*[Response]* Thank you for your comment. The latest results of dust simulation using the WRF model (Zhao et al., 2020; Shukla et al., 2021) and details about what new has been done in this manuscript were added in manuscript. Besides, in introduction part, the citations from Chen and Hu mentioned on line 49 and line 63 used WRF-Chem to simulate dust, which have a high number of citations.

"Zhao (Zhao et al., 2020) used WRF-Chem to simulate a dust process in Northwest China during May 2018 and compared results between the simulations based on five dust emission schemes within WRF-Chem and the observations show that, WRF-chem has good performance in simulating the emission flux, the spatial pattern of source region, as well as the spatiotemporal variation of dust mass concentration, except Shao 11 scheme. Shukla (Shukla et al., 2021) used a high-resolution WRF-Chem to analyze the dust storm that occurred during 12 17 June 2018 over the northwest Indo-Gangetic Plain, the horizontal and vertical distributions of dust aerosols reproduced by the WRF-Chem agree with the observations and the evolution of dust storm and associated changes in atmospheric and air quality conditions were well simulated."

"A quantitative study of the spatial and temporal variability of dust fluxes and atmospheric dust properties of different climatic regions in the interior of the plateau was conducted using WRF-Chem in three regions over a three-year period (2004-2006) with high resolution to see the finer dust source distribution and dust characteristics on the plateau under different climate zones."

**2: Line 65: "from the northern part of the plateau to the plateau". What does the authors mean here? Please explain.**

*[Response]* Thank you. This error has been corrected.

"Hu (Hu et al., 2020) studied the total amount of dust transported to the TP from different dust sources at different altitudes, and dust from North Africa and East Asia was primarily transported to the northern part of the plateau and from the Middle East was transported southward and lifted up to the TP."

**3: Line 82: "there are a few studies on dust inner the plateau". May be this sentence needs restructuring.**

*[Response]* Thank you. This error has been corrected.

"At present, there are a few studies focused on dust aerosols within the plateau, and conclusions regarding the distribution of dust sources inner the plateau, the spatial and temporal variability of dust initiation, and the contribution of dust to the air column above the TP are still controversial."

**Study region and Data**

**4: Line 114: "Therefore, we divided the study area into three regions according to climatic characteristics (Figure 1)." The previous lines have references Zheng, 1996; Xu et al., 2013. Which one of these does the authors have used for "climatic characteristic". If some other reference is used, then please mention here.**

*[Response]* Actually, the literature of Zheng was used for "climatic characteristic". And the literature of Xv was used to demonstrate that the distribution of precipitation and vegetation on the plateau is in good agreement with climatic zonation, which further

proves the accuracy of regional division.

**5: The methodology used to perform the WRF simulations has not been explained well in this manuscript. The authors provide a brief description of the schemes that they have used in Section 2.2.1 But they have not provided any details on the on-line coupling of the WRF-Chem model as they mention in line 123. The authors are suggested to provide a detail on the simulations performed and how the on-line coupling of meteorology and chemistry is performed. Also, the authors should provide details on how the simulations are performed for the three different regions mentioned in Section 2.1. Are they performed independent of each other or through nested domains?? If the simulations are performed independently for each domain, then the authors should provide a reasoning for this i.e., if the boundary conditions will be accurately represented for each domain. If nested domains are used, then the authors should provide details of these simulations.**

*[Response]* Thank you for your comment. A detail on the simulations performed and how the on-line coupling of meteorology and chemistry is performed and details on how the simulations are performed for the three different regions mentioned in Section 2.1 were provided.

"The WRF-Chem model (V3.9.1) is based on the WRF model with the introduction of the Chem module and full on-line coupling of meteorology and chemistry, the chemistry package consists of dry deposition (''flux-resistance'' method), biogenic emission, the chemical mechanism from RADM2, a complex photolysis scheme (Madronich scheme coupled with hydrometeors), and a state of the art aerosol module (MADE/SORGAM aerosol parameterization). (Grell et al., 2005). In this study, the simulated area had a horizontal resolution of 6 km, with $99 \times 99$ grid points in the west, $99 \times 334$ grid points in the south and $115 \times 334$ grid points in the north. Each area was simulated separately to better reflect local dust emission and loading characteristics without allowing them to interfere with each other. The depth from the surface to 50 hPa was vertically divided into 40. The simulation process began on January 1, 2004, with monthly updates of the initial field to avoid bias in the boundary conditions and

initial fields due to long simulation times, and 24 h after 00:00 on the last day of the previous month was used to allow the model to spin up. The boundary and initial conditions were obtained from FNL global reanalysis data provided by the National Centers for Environmental Prediction at a resolution of 1 °, which were available every 6 h. The model was output in netcdf format at half-hour intervals. The following parameters were chosen for the simulation: Purdue Lin microphysical scheme (Gustafson et al., 2007), Yonsei University planetary boundary layer scheme (Noh et al., 2006), RRTMG long-wave and short-wave radiation scheme (Mlawer et al., 1997; Iacono et al., 2000), Noah scheme (Chen et al., 1996), and Grell cumulus scheme (Grell et al., 1994); other parameter settings refer to Chen (Chen et al., 2013) and Zhao (Zhao et al., 2020).

In GOCART model all topographic lows with bare ground surface are assumed to have accumulated sediments which are potential dust sources, and the uplifting of dust particles for each grain size is expressed as a function of surface wind speed and wetness. (Ginoux et al., 2001), and the dust emission flux equation is calculated as follows:

$$G=CSs_{p}U_{10}^{2}\ (U_{10}\text{-}U_{t})$$

where $C$ ($\mu gs^2m^{-5}$) is a constant; $S$ is a source function that defines the area of potential dust sources, consisting of vegetation, snow, and other surface factors; $s_p$ is the proportion of dust emissions for the different classifications. Dust was classified into five classes in the GOCART scheme, with particle size intervals of 0.2-2, 2-3.6, 3.6-6, 6-12 and 12-20 μm. $U_{10}$ is the horizontal wind speed at 10 m above ground level. $U_t$ is the threshold wind speed; when the wind speed is less than this value, the dust will not be blown up into the atmosphere. The threshold wind speed is a function of particle size, air density, and surface soil moisture."

**Model evaluation:**

**6: Line 170: "three regional models". What does the author mean by "three regional models"?? The authors are only using WRF-Chem model in this manuscript. May be the authors are referring to the three regions of the model. If not, then provide an**

explanation here. Please rephrase this sentence to avoid confusion.

*[Response]* Thank you. This error has been corrected.

"The model results of the three regions were gridded and averaged to compare the annual variation in the regional average characteristics with the corresponding areas in the reanalysis data."

**7: Line 175: "The model results were slightly lower than the reanalysis results.". This means that the model is underpredicting these values as comapred to the reanalysis values. Can the authors provide a probable reason for this underprediction??**

*[Response]* Thank you for your comment. There are a number of reasons why the simulation results are lower than the reanalysis results. A few possible reasons should be taken into consideration, including incapability of the model to simulate all the thermal and dynamic processes in the real world. In particular, the Tibetan Plateau has large topography and complex underlying surface, the mechanisms involved are more complex. Therefore, the simulation results cannot be exactly the same as the observation results and the reanalysis results.

Besides, many studies have compared the various reanalysis data with the observations and concluded that the MERRA-2 reanalysis data is the most consistent with the observations (Carvalho., 2019. https://journals.ametsoc.org/view/journals/clim/32/23/jcli-d-19-0199.1.xml). The reanalysis data can make up for the lack of observation data in the area of harsh natural environment, such as the Tibetan Plateau. In Figure 4, we also compared the results of observation data, reanalysis data, and model result in the southern region, it shows that the results of three kinds of data are consistent in the annual trend, but there are slight differences in the value. The relative error of temperature above 2 m between simulation results and reanalysis data is 0.8% (west), 0.7% (south), and 0.1% (north), this indicates that the simulation results are satisfactory.

What's more, through literature reading, we found that in many studies, there were small numerical errors between the reanalysis data and the model results when the model was verified, and generally the simulation results are smaller than the reanalysis

data (Chen et al., 2013; Zhao et al., 2021). As described above, such errors are unavoidable due to technical reasons.

**8: Line 179: "The model results and reanalysis data showed a significant difference in wind speed in summer, with the model results being significantly lower than the reanalysis data, with a difference of 1 m/s". Can the authors provide an explanation for this finding?**

*[Response]* Thank you for your comment. The main explanation was provided at #7. And the relative error of wind above 10 m between simulation results and reanalysis data is 3.3% (west), 21.8% (south), and 22.8% (north). In the simulation results, temperature is the closest to the observation results, followed by wind speed, and the worst is precipitation (Clark and Gerhard, 2017 https://www.tandfonline.com/doi/full/10.1080/07055900.2017.1282345), so we think the simulation results are satisfactory.

**9: Figure 3: There is something wrong with the legend of this figure. What does the dashed line and the solid lines represent?? The legend only has description of solid lines.**
*[Response]* Thank you. This error has been corrected.

**Results:**
**10: Line 248: "maximum dust emission in January". Can authors explain the reason why the dust emission was maximum in a winter month of January and it was lower in summer months, in fact minimum in summer month of August. Why is the emission maximum in Winter and not in summer?? If the frequency of dust storms are more in winter months, then is it possible that the maximum dust emission values are arising due to the frequency, but is it possible that the dust emissions are more in summers but when averaged for the month, then the values comes out to be less?? If so, then has the authors take the frequency of these events into account?? Also, these emission are at what level (at the surface or the averaged values over the 40 vertical layers used in the simulation)??**

In line 252, the authors mention "This annual trend was consistent with the observed annual trend in dust storm frequency". But has the frequency of occurrence of these events taken into account and considered while averaging the emissions for a month?

*[Response]* Thank you for your comment. Figure 3 shows that the maximum wind speed at 10 m occurring in January, which mostly explained why the dust emission was maximum in a winter month of January and it was lower in summer months. In Tibetan Plateau, the precipitation in winter is less than that in summer, and the soil moisture is lower. Dry underlying surface conditions will reduce the threshold wind speed (Yang et al., 2019 https://doi.org/10.1007/s11069-019-03686-1). High wind speeds in winter are more likely to reach the wind speed threshold and more likely to blow dust off the ground in TP. As described in Section 2.2.1, the uplifting of dust particles for each grain size is expressed as a function of surface wind speed and wetness.

Actually, the amount of dust emissions we calculate is the total amount of dust raised per month, that means accumulation of dust emissions of all dust events in a month. So it may be that there are more dust storms in winter but the amount of dust emissions each time is small but that doesn't affect the fact that in terms of the monthly total, the total amount of dust emissions in the three months of winter is higher than in summer. However, it is a good inspiration for the following work to analyze the total amount of dust emissions in a single process of winter and summer, respectively.

"This trend of higher winter and lower summer is also influenced by soil moisture, dry soil conditions in winter reduce the wind speed threshold, making it easier for dust to be lifted into the atmosphere at the same wind speed (Yang et al., 2019)."

**11: Line 263: "Wu (Wu et al., 2019) used multiple models to simulate the annual variation in the multiyear average optical thickness of dust over the plateau; they also showed a similar bimodal structure.". In this work by Wu et. al., authors have presented that for TP, the dust emission is more in March-April than in January. While in this (current manuscript) work, the authors have presented that the emissions are maximum in January and then decreases to a minimum in August. Can the authors support this claim??? Also, which bimodal structure are the authors talking about??? Please explain**

this.

*[Response]* Thank you for your comment. In fact, Wu compared the results of several models simulating the characteristics of plateau dust. What you are talking about is only one model (K14_f09) and the other models all show a gradual decline in the maximum in January to a low value in the summer and then a slow recovery. Perhaps the numerical results of other models are smaller than that of K14_f09 model, making them not stand out.

The bimodal structure refers to the occurrence of two peaks of DOD in April and June or June and August, or April and August.

**12: Figure 7: I assume that the solid lines represent the model simulation in figure 3, then in North region the simulated temperature is higher from April to September, while wind speed remains mostly constant. Figure 6 shows that the dust emission is low in these months. Can the authors explain why the dust loading in North region is quite high from April to September (Figure 7).**

*[Response]* Thank you for your comment. The reasons of the dust loading in North region is quite high from April to September are listed as following:

1. The most important reason is that the unique topographic features of the northern region. The Qaidam Basin of northern region is not a high-value area for dust emissions in winter, but in summer. The impact of unique basin topography on a cold frontal system, a large amount of cold air invading the TP in three main tracks of near-surface winds, which allowing dust to remain over the plateau for longer (Meng et al., 2019 https://www.sciencedirect.com/science/article/pii/S0169809518306410?via%3Dihub)

2. The role of heat pumps in the Tibetan plateau. The summer heat effect enhances the transport of dust to higher altitudes, allow more of the dust to reach the upper troposphere and remain in the atmosphere. However, in the southern region, with frequent summer rains and lush vegetation, it is difficult for dust to stay in the atmosphere for a long time.

"The main reason make the dust loading of northern region quit higher in summer is that the unique topographic features of the northern region. The Qaidam Basin of northern region is not a high-value area for dust emissions in winter, but in summer. The impact of unique basin topography on a cold frontal system, a large amount of cold air invading the TP in three main tracks of near-surface winds, which allowing dust to remain over the plateau for longer (Meng et al., 2019)"

"The southern region is wet year-round, with lush ground vegetation and snow-covered mountains, which make dusts difficult to stay in the atmosphere for a long time; therefore, there are more factors influencing dust deposition and dispersion and the correlation is low."

**13: Figure 8: How is the explanation provided for Figure 8 related to Figure 7? It would be nice if the authors can link these two figure along with a proper explanation. *[Response]* Thank you for your comment. The description of Figure 8 is not clear enough, the detailed description and additional content has been added to the article.**

"Therefore, we calculated the proportion of dust above 8km to the total dust content of the whole air column to quantitatively calculate how much of the dust suspended in the atmosphere can be transported to the upper troposphere and further outwards (Figure 8)."

---

## Author Comment (AC3)

This manuscript reports results of WRF model simulation to quantify the spatial and temporal variability of dust emissions, dust loading, and their characteristics within the Tibetan Plateau during 2004-2006. The authors divide the main body of the Tibetan Plateau into three regions, and argue that the results of the simulations about dust activity were consistent with the reanalysis data and ground observations.

Overall, simulation of winter dust on the Tibetan Plateau is with great importance. However, this manuscript has some shortages, and some of the implications and discussions in the manuscript are not appropriate. I have a number of comments and concerns listed below, and the authors should make Major revisions of this manuscript before it can be potential published on ACPD.

*[Response]* Thank you very much for your suggestion. In particular, many references are provided that allows me to verify the research results and modify next steps of my research plan. We have taken all of the specific comments into account in the revised version of the manuscript. Please see the detailed response marked blue below and the changes marked red in the revised manuscript.

**Main comments**

1.The current definition of the Tibetan Plateau is confusing and not appropriate. In common sense, the Tibetan Plateau is the region higher than 2500 or 3000m above sea level. The western of the Sichuan Basin, the western of the Loess Plateau, the eastern of the Tarim Basin and other regions shown in Fig. 1 cannot be incorporate into the Tibetan Plateau. Please read the definition by Yao et al., 2012, Nature Climate Change (DOI: 10.1038/NCLIMATE1580) and Liu et al., 2022, Global andPlanetary Change (https://doi.org/10.1016/j.gloplacha.2022.103893).

*[Response]* Thank you for your comment. In this manuscript, which definition of the Tibetan Plateau we chose is that the region extends from the Pamir Plateau in the west to the Hengduan Mountains in the east, from the southern edge of the Himalayas in the south to the northern side of the Kunlun-Qilian Mountains in the north. Li Bingyuan (1987) discussed the principles and specific boundaries for determining the extent of the Tibetan Plateau in a more systematic manner, and proposed that the basic principles for determining the extent of the plateau should be based on the geomorphological features, the plateau surface and its altitude, while taking into account the integrity of the mountains. Zhang Yili

(2002) argued for the principles of determining the extent and boundaries of the Tibetan Plateau based on new research results in related fields and years of field practice, and combined with information technology methods to precisely locate and quantitatively analyze the extent and location of the boundaries of the Tibetan Plateau, then they put forward this definition of the Tibetan Plateau. ( http://data.tpdc.ac.cn/zh-hans/data/61701a2b-31e5-41bf-b0a3-607c2a9bd3b3/)

Actually, the region higher than 2500 or 3000m above sea level was widely used by many scholars, but we chose the definition mentioned above because we want to cover the main body of the Tibetan Plateau as much as possible, meanwhile avoiding the influence of large drop in elevation. Besides, WRF-Chem can only provide regular rectangular areas for simulation, grid point with a resolution of 6 km is the compromise between resolution and simulated regions so that there are much more other regions were considered in simulation. Therefore, the difference of area between two definitions on this region could be ignored compared with the giant body of the Tibetan Plateau.

2. The division of the three regions on the Tibetan Plateau is crude. The authors mentioned that the resolution of WRF model was 6 km, it can be more precise for division of those three regions. The dust sources of the plateau are located in the northern part of the Qiangtang Plateau, the Yarlung Tsangpo River basin, the Namucuo and Lhasa regions, the Qaidam Basin, the source areas of the Yellow and Yangtze Rivers, and the Qinghai Lake and its surrounding areas. Therefore, Istrongly suggest the authors use Qiangtang Plateau (between the Kunlun Mountains and Gangdisi Mountains) as the western Tibetan Plateau, north of the Tanggula Mountains (roughly around 32-33 °N) as the Northern Tibetan Plateau, and south of the Gangdisi Mountains-Tanggula Mountains as the Southern Tibetan Plateau, on the basis of the definition of the Tibetan Plateau (higher than 3000m or 2500 m).

*[Response]* Thank you for your comment. There has been controversy over the division of the three regions on the Tibetan Plateau. The original intention in designing the experiment is that we want to ensure the remote inland in the western Tibetan Plateau is a source of sand and dust (such as Ali and its surrounding areas) and make a quantitative description of dust emissions in this area. Our team have carried out long-term aerosol investigation and observation experiments in the Ali region (Zhang et al., 2021, https://doi.org/10.1029/2020JD033286), during this time we found that Small dust events occur frequently in Ali and its surrounding areas. So when we divided regions, we tried to cover these areas and exclude surrounding deserts. As for the northern region and southern region we chose the 85 °E and 33 °N as the boundary because of the difference of temperature and precipitation. (Jiang and Wang,

2001, Jia et al., 2015).

But combined with the results of this research, the division of the three regions on the Tibetan Plateau from you could be as a good reference for the delineation of future research work areas. Thank you again.

3. The authors only provide results of three years during 2004-2006. However, a climate pattern (30 years, such as 1991-2020) can provide more confident results and reliable conclusions.

*[Response]* Thank you for your comment. Wang and Zhang (2006) analysed the number of days occurred with dust and sandstorm weather (including floating dust, sand, dust storms and strong dust storms) based on observations from meteorological stations. The results show that there are two main areas of high dust and sandstorm weather occurrence in Northeast Asia during spring 2006, one occurring in the Taklamakan Desert and surrounding areas in southern Xingjiang, China, and the other being probed in the arid region of southeastern Mongolia, (https://kns.cnki.net/kcms/detail/detail.aspx?dbcode=CPFD&dbname=CPFD9908&filename=ZGQX 200610001000&uniplatform=NZKPT&v=AmhKSmhN9Hr4mkIG2ftYQjIhYCFvE0I3ZZDE2M8SR 2XxbFxuQzrQEE90QS_Owhk16TIlsFxDV78%3d). And a dust storm was observed on 22, Jan, 2006. So we simulate the Tibetan Plateau from 2004 to 2006 to quantify the spatial and temporal variability of dust within the plateau.

Obviously, a long duration simulation is important, but short simulations are also indispensable. Results of long simulations are prone to bias, Short-time simulations do not reflect trends. The results of this short simulations proved that this way to simulate dust emissions of the Tibetan Plateau is credible. And Long-term simulation about the Tibetan Plateau dust emissions will be my next work plan.

4. The dust emission rate data on the Tibetan Plateau throughout the manuscript should be checked carefully. Previous studies reported that dust emission rate in the Taklimakan desert was about 0.38 ton/ha yr (equals to 38 $g/m^2$), in the Central gobi-desert is about 0.24 ton/ha yr (equals to 24 $g/m^2$) (see Xuan et al., 2002, Atmospheric Environment, https://doi.org/10.1016/S1352-2310(02)00585-X). In comparison, the authors report dust emission rates of about $11.00 \times 10^7 \, \mu g \cdot m^{-2}$ (equals to 110 $g/m^2$) in

the west, $3.30 \times 10_7$ μg·m$^{-2}$ (equals to 33 g/m$^2$ ) in the south and $4.5 \times 10_7$ (equals to 45 g/m$^2$ ) in the north during winter on the Tibetan Plateau, in this work. I am not sure whether dust emission rate on the Tibetan Plateau can be much higher than the Taklimakan desert and Go-bi desert?

*[Response]* Thank you for the reference that you recommended, I have carefully investigated this article and discussed the details with my co-authors. In this article, the authors use 30-year (1951–1980) climatological data and the modified US EPA empirical formulas to finish the research. The result that dust emission rate in the Taklimakan desert was about 0.38 ton/ha yr (equals to 38 g/m$^2$ ), in the Central gobi-desert is about 0.24 ton/ha yr (equals to 24 g/m2 ) is averaged over many years, it may have smoothed out some great values. Besides, the data between the reference and us used to simulate spanned a large time, from 1980 to 2004. And different models have various schemes of dynamic and thermal mechanism. So it is normal to see different results.

According to the latest research that the observations and simulated results showed that the QTP had a high dust emission rate, and the dust (PM10) emission intensity in the QTP during the period of 2000 to 2020 ranged from 0 to 1042 g/m$^2$/y with a mean value of 32.4 g/m$^2$/y. (Du et al., 2022, https://www.sciencedirect.com/science/article/pii/S0016706122002373). And the averaged value from our simulations is 910 g/m$^2$/y, So I think the result is credible.

5. In the abstract, "few studies have been conducted on dust aerosols within the plateau", and Line 80-83, "there are few studies on dust within the plateau, and conclusions regarding the distribution of dust sources within the plateau, the spatial and temporal variability of dust initiation, and the contribution of dust to the air column above the TP are not clear". As far as I know, there are many studies on dust aerosols in the TP. The sentences are too absolute. For example, Liu et al., ACP 2008; Kang et al., AE 2016; Mao et al., SCIENCE CHINA Earth Sciences 2013;Mao et al., AE 2019; Yuan et al., JC 2019, etc. Mao et al., did a lot of work on dust using WRF-Chem in the TP. What are the differences between your manuscript and Mao's work? The authors should highlight your characteristics and differences relative to previous studies. In addition, please add the dynamic mechanism related to your results in the abstract.

*[Response]* Thank you for the references that you listed, I have investigated these articles carefully, and the text has been revised accordingly. Obviously, Mao's work has led to a number of important results and has provided much assistance in the design of the research work and validation of the results

in this paper, this article also cites many of the results of his work (Mao et al., 2013, Feng et al., 2020). But they analyzed the Tibetan Plateau as a whole in their simulations and chose a coarser grid resolution.

As stated in the introduction to our article, the complex topography and sub-surface conditions make the climate very different in different regions of the TP. So the characters of dust emissions and dust loading in sub-regions of the TP are significantly different. However, setting and analyzing the Tibetan Plateau as unity in simulation only reflects northern character only, because of the high value of dust loading of northern region. There are two pictures below, left one is dust optical depth (Wu et al., 2019, https://agupubs.onlinelibrary.wiley.com/doi/full/10.1029/2019JD030799), right one is dust loading of this article, they show that both the trend of dust optical depth in the Tibetan plateau and the trend of dust loading in the northern region are bimodal structure, but the trend of dust loading in the western and southern region are not. So sub-regional study of the Tibetan Plateau could be useful for future studies on the climatic effects of the Tibetan Plateau aerosols.

What's more, finer resolution can better reflect the details of the spatial distribution of dust on the Tibetan Plateau. And the abstract and introduction were modified.

[revised manuscript text omitted]

6. If the authors add the model evaluation of AOD between the results of WRF and MODIS or MISR, cross sections of dust extinction coefficient between the results of WRF and CALIPSO, the results will be more reliable.

*[Response]* Thank you for your comment. According to annual variation in dust loading in three regions on the Tibetan Plateau (Figure 7), we compared data of average monthly AOD between the results of WRF and MODIS on April as following. They show that the simulation results were consistent with the spatial distribution of the MODIS data,but numerically lower than MODIS.

However, this article focused on the dust in the interior of the Tibetan Plateau while MODIS observe dust from global sources. Therefore, although comparison of the results verifies that the simulation data is reliable, this part of the result is not added to the article. If necessary, we will upload it to supplement.

[Figure]

7. The language of this manuscript needs editing by a native speaker.

*[Response]* Thank you. The language of this manuscript was edited by a native speaker.

**Other comments**

1. Line 36, the definition of the Tibetan Plateau

*[Response]* Thank you*.* The definition of the Tibetan Plateau has been introduction at main comment 1. And "The Tibetan Plateau (TP) is the highest-elevated landmark in the world, with elevations ranging from approximately 1500 to 5000 m above sea level." Can be found in Liu et al., 2019.

2. Line 45, Batangilin Desert? Is it Badanjilin Desert/ Badain Jaran Desert?

*[Response]* Thank you. This error has been corrected

"There are many well-known dust source areas around the TP, such as the Taklamakan Desert, Thar Desert of India, Badain Jaran Desert, and Gurbantung Desert."

3. Line 53, the unit of 6.6G is "g" or "ton"?

*[Response]* Thank you. This error has been corrected

"This caused the dust to breake through the planetary boundary layer and extend into the upper troposphere in the northern part of the plateau and produced an average daily transport of 6.6 Gg of dust from the Taklamakan Desert to the TP over the 5 days of the event."

4. Lines 79-82, the inner dust sources on the Tibetan Plateau are clear, please read Wu et al., 2013, Quaternary Science Reviews (http://dx.doi.org/10.1016/j.quascirev.2012.10.003).

*[Response]* Thank you for the references that you listed, I have studied these article carefully, this article present the detailed dust history from 1850 to 2004 AD on an annual timescale from a shallow ice core from Tanggula, central Tibetan Plateau. The analysis leads to the conclusion that Tarim Basin and over the western Tibetan Plateau, are possible source areas. They also ensure that dust come from Udaoliang made up the largest proportion of the ice core record period. Tanggula Mountains are so far and high-elevated to be the representative of inland of the Tibetan Plateau, and the reestablishment of dust spatiotemporal distribution could be promoted with corporation between numerical simulation and climatological records, such as ice core. However, the dust sources and relative importance are still controversial, there are a lot of dust that cannot be transported to the Tanggula Mountains.

"At present, there are a few studies on dust inner the plateau, and conclusions regarding the distribution of dust sources inner the plateau, the spatial and temporal variability of dust initiation, and the contribution of dust to the air column above the TP are still controversial."

5. Line 93, similar to former major concern, why only choose 2004-2006?

*[Response]* Thank you. The definition of the Tibetan Plateau has been introduction at main comment 3.

6. Lines 101-102, I strongly suggest the authors provide a map of sand dunes and deserted land on the Tibetan Plateau in the manuscript or in the supporting information

*[Response]* Thank you. The map of sand dunes and deserted land on the Tibetan Plateau can be found in Fang, 2004; Han et al., 2009.

"The widespread sand dunes and deserted land on the TP provide a large amount of raw material for the generation of dust storms(Fang, 2004; Han et al., 2009)"

[Figure]

[Figure]

Fig. 1. Distribution of the modern and last glacial sand dunes, desertified sandy land, and the mean annual dust storm days and precipitation from 1961 to 2000 on the TP (black solid line: precipitation; black dashed: dust storm days; black circle including the number representing the following locations: 1, Guliya ice core; 2, Malan ice core; 3, Dunde ice core; 4, Delingha tree ring).

[Figure]

Fig. 1. The distribution map of modern and last glacial sand dunes, desertified sandy land and loess and the spatial mean annual days plot of dust storm from 1961 to 2000 on the Tibetan Plateau. Aeolian sand and loess distribution was drawn according to our field investigation and refs. [1~5], [9~12] and [19~21]. Dust storm data are from the Climate Center of China Meteorological Administration.

7. Lines 102-115, the climate patterns are the basis for the division of the three regions on the Tibetan Plateau. The authors should read some most recent papers about the climate modes on the Tibetan Plateau, redefine the three regions and rewrite this paragraph.

*[Response]* Thank you. The definition of the Tibetan Plateau has been introduction at main comment 2.

8. Fig 1. The north and south are wrongly labeled, the authors should better add the variation of meteorological parameters such as temperature and precipitation of these three regions.

*[Response]* Thank you. This error has been corrected, the variation of meteorological parameters such as temperature and precipitation of these three regions can be found in Jiang and Wang, 2001.

9. Line 206, Gondola Mountains?

*[Response]* Thank you. This error has been corrected

"except for the area where the Gangdise Mountains are located"

10. Lines 236-236, are you sure that the modern East Asian summer monsoon can reach the Qaidam Basin and cause heavy rainfall? If yes, please cite the suitable references. To our knowledge, the maximum of the East Asian summer monsoon is around Lenglongling in the eastern Qilian Mountains.

*[Response]* Thank you. The suitable reference was cited. Xv et al., 2011 (https://agupubs.onlinelibrary.wiley.com/doi/full/10.1029/2010JD015053)

"In summer, the Qaidam Basin is primarily affected by the East Asian summer wind, and local heavy rainfall, hail, and other strong convective weather are prone to occur, thus causing local windy weather (Zhang et al., 2014; Xu et al., 2011)."

11. Lines 271, "the dust in the north stayed in the atmosphere for a long period of time", does this mean the potential effects of dust from the Tarim basin?

*[Response]* Thank you. In fact, some of the dust over the northern region indeed came from the Tarim Basin. But the Figure 5 in this article shows that there are many dust sources in the northern region. Compared with western region, more man-made events are reported here. And compared with southern region, precipitation is less here. These reasons make the dust in the north stayed in the atmosphere for a long period of time

12. Lines 289-291, the western Tibetan Plateau is mainly the Qiangtang Basin, the landscape consists planation surface

*[Response]* Thank you. Our team has carried out long-term aerosol sampling and observation experiments in the Ali region (Zhang et al., 2021, https://doi.org/10.1029/2020JD033286). At present, our team are still sampling around the Tibetan Plateau. They did observe sand dunes and permafrost.

[Figure]

[Figure]

13. Lines 304-305, the height 8 km above sea level reaches the middle-upper troposphere, does this meant the height of the boundary layer over the Tibetan Plateau was about 3-4 km above the ground level?

*[Response]* Thank you. The height of 8 km is come from Chen et al. 2013. We chose the highest altitude to be the dividing line in order to ensure dust that once overpasses it can't drop back to the ground.

[Figure]

**Figure 9.** (a–d) Vertical-latitude cross-section at 85.5°E of daily averaged aerosol extinction coefficient and potential temperature on 26–29 July from the WRF-Chem simulations over domain 1. Black contours indicate the potential temperature. The red line represents the daily averaged planetary boundary layer height (PBLH).

14. Please add the linkages of the meteorological and other dataset used in this paper in the Acknowledgement section.

*[Response]* Thank you. The linkages of the meteorological and other dataset used in this paper in the Acknowledgement section were added.

"We thank to the MERRA-2 reanalysis data were provided by NASA (https://disc.gsfc.nasa.gov/) and the wind speed and temperature observations from 19 stations in the Tibetan region provided by the China Meteorological Data Network (http://data.cma.cn/). We also thank the Land cover products (https://zenodo.org/record/4280923#.Yzg4dmu-vqA). The authors acknowledge support from the Second Tibetan Plateau Scientific Expedition and Research Program (STEP) (No. 2019QZKK0602) and support from Supercomputing Center of Lanzhou University. We also acknowledge the editor Stelios Kazadzis and two referees."

15. References should be updated. For instance, line 489 "Journal of geophysical research. Atmospheres: JGR, 124, 8043-8064" is not correct.

*[Response]* Thank you. This error has been corrected

"Wu, M., Liu, X., Yang, K., Luo, T., Wang, Z., Wu, C., Zhang, K., Yu, H., and Darmenov, A.: Modeling Dust in East Asia by CESM and Sources of Biases, Journal of geophysical research: Atmospheres, 124, 8043-8064, 10.1029/2019JD030799, 2019."

16. Please check all the reference format. Like in line 55, 61, "the results of Wang(Wang et al., 2021)" , "Hu(Hu et al., 2020)", not correct.

*[Response]* Thank you. This error has been corrected